# Sleep Disorders in a Sample of Patients with Pediatric-Onset Multiple Sclerosis: Focus on Restless Legs Syndrome

**DOI:** 10.3390/jcm14093157

**Published:** 2025-05-02

**Authors:** Elena Panella, Laura Papetti, Martina Proietti Checchi, Samuela Tarantino, Michela Ada Noris Ferilli, Gabriele Monte, Alessandra Voci, Claudia Ruscitto, Luigi Mazzone, Massimiliano Valeriani, Romina Moavero

**Affiliations:** 1Child Neurology and Psychiatry Unit, Systems Medicine Department, Tor Vergata University of Rome, 00133 Rome, Italy; 2Developmental Neurology Unit, Bambino Gesù Children’s Hospital, IRCCS, 00165 Rome, Italy; 3Translational Pain Neuroscience and Precision Medicine, Center for Neuroplasticity and Pain, Department of Health Science and Technology, School of Medicine, Aalborg University, 9220 Alborg, Denmark

**Keywords:** sleep disorders, restless legs syndrome, restless sleep disorder, pediatric-onset multiple sclerosis

## Abstract

**Background/Objectives**: Sleep disorders (SDs) and Restless Legs Syndrome (RLS) have been reported with high prevalence in Multiple Sclerosis (MS), but data on Pediatric-Onset MS (POMS) are scarce. This study aims to assess the prevalence of SDs, particularly RLS, in a POMS cohort and examine associated clinical features. **Methods**: We recruited POMS patients who attended the POMS Center of the Bambino Gesù Children’s Hospital between September 2021 and February 2023; they were evaluated for SDs using the Pittsburgh Sleep Quality Index (PSQI) or the Sleep Disturbance Scale for Children (SDSC) and screened for RLS. Correlations with demographical, clinical, neuroradiological, and laboratory findings were analyzed. **Results:** We recruited 44 POMS patients, of whom 39% were classified as “good sleepers” and 61% were identified as “poor sleepers.” RLS was diagnosed in 10 patients (22.7%). Those with RLS were older and had higher Expanded Disability Status Scale (EDSS) scores compared to non-RLS patients (*p* = 0.028; *p* = 0.03). The presence of RLS did not show any significant correlation with MRI lesion load or laboratory data. **Conclusions**: Our findings suggest an increased rate of SDs and RLS in pediatric MS patients compared to the general pediatric population. Clinical data could support a secondary form of RLS in this population, but results need further confirmation.

## 1. Introduction

Multiple Sclerosis (MS) is a chronic neurological disease that affects the Central Nervous System (CNS), typically manifesting between the ages of 20 and 40 [1]. Less commonly, MS begins before the age of 18, accounting for approximately 3–5% of cases; this form is referred to as Pediatric-Onset Multiple Sclerosis (POMS) [2]. The exact pathophysiology of MS remains unclear, but it is widely believed to involve an autoimmune response triggered by environmental factors in genetically predisposed individuals. This response leads to inflammatory infiltration in the CNS via the blood-brain barrier, resulting in demyelination, gliosis, and neuronal damage [2]. In pediatric patients, the most common manifestation of MS is Relapsing Remitting MS (RRMS), characterized by episodes of clinical relapses with CNS symptoms alternating with phases of remission. Compared to adult-onset MS, POMS tends to have a more inflammatory and active disease course, with a higher relapse rate, particularly in the initial years following diagnosis [2]. Despite variability in clinical manifestations, the most frequent symptoms of POMS include motor deficits (30%), sensory disturbances (15–30%), brainstem symptoms (e.g., diplopia, facial sensory symptoms, facial weakness) (25%), optic neuritis (10–22%), and ataxia (5–15%) [2]. POMS is also commonly associated with fatigue (20–43% [3]), depression (17–29% [4]), non-motor MS symptoms (31% [5]), and sleep disturbances (42–65% [6] in adults, with no defined prevalence in pediatric patients [7]). All these factors significantly impact patients’ quality of life [4]. POMS patients report a lower quality of life compared to the general population, particularly in areas related to school, social engagement, and emotional well-being [4,8].

Sleep disorders (SDs) are a significant concern in MS, impacting both physical and mental health, especially in chronic conditions. While SDs are well-recognized in adults with MS, their prevalence in POMS remains less well-established [6,9]. However, the impact of sleep on quality of life, fatigue, and physical activity is already acknowledged in this population [7]. Additionally, sleep disturbances can negatively affect mental health, leading to mood swings, behavioral changes, and social difficulties. This suggests that sleep patterns should be carefully examined to improve the well-being of POMS patients [7,10]. Common SDs in POMS include insomnia, sleep-related movement disorders, sleep-related breathing disorders, and circadian rhythm disorders [9]. Considering the high prevalence of bladder dysfunctions in POMS, we should consider that nocturia, defined as the need to wake up to urinate during the night, could affect sleep quality, although no specific data are currently available on the pediatric population [11,12].

RLS, categorized among sleep-related movement disorders [13], is characterized by an urge to move the legs, worsening in the evening, often accompanied by unpleasant sensations, and relieved by movement or walking. Diagnostic criteria, established by the International Restless Legs Syndrome Study Group (IRLSSG) [14] and adapted for pediatric patients [15] (Table 1), include five key criteria.

RLS significantly impacts sleep quality, daily functioning, and mood [16], but remains underdiagnosed, affecting 5–13% of the population in Europe and North America [16,17]. It can be primary or secondary, often linked to iron deficiency, pregnancy, renal failure, and neurological disorders such as MS and Parkinson’s disease [18]. Though once thought to affect only middle-aged and older adults, RLS is now recognized in children, with a prevalence of 2–4% [17,18]. Risk factors include Periodic Limb Movement Disorder and a family history of RLS [19]. The incidence of RLS in MS patients ranges between 13.5% and 65% [20], much higher than in the general population, and correlates with longer disease duration and greater disability [20,21]. Only one study has specifically investigated RLS in POMS, showing a similar increased prevalence and its association with higher disability levels [22].

Additionally, the IRLSSG recently identified Restless Sleep Disorder (RSD), characterized by sleep disruption and frequent nocturnal body movements involving large muscle groups, causing daytime impairment [23]. Unlike RLS, RSD requires polysomnography for diagnosis (Table 2).

While its prevalence in the general population is unknown, RSD has been found in 7.7% of children with sleep concerns [24]. The pathogenesis of RSD is not well defined, though it may share some risk factors with RLS, such as iron deficiency [25].

This study aims to assess the prevalence and characteristics of sleep disorders, particularly RLS, in our cohort of patients with POMS. We also aimed to describe their demographic, clinical, neuroradiological, and laboratory features, and to evaluate a possible correlation with quality of life.

## 2. Materials and Methods

### 2.1. Subject Recruitment

This cross-sectional study was conducted at the POMS Center of the Bambino Gesù Children’s Hospital (Developmental Neurology Unit). All patients with POMS followed in our clinic were consecutively recruited between September 2021 and February 2023. Inclusion criteria were as follows: (i) a diagnosis of MS before the age of 18, based on the criteria of the International Pediatric MS Study Group (IPMSSG); (ii) any relapse events occurring 30 or more days prior to recruitment; (iii) availability and ability to respond to the questionnaires. Patients with comorbid conditions, or those using medications that could affect sleep (e.g., benzodiazepines, antidepressants, antihistamines, steroids) were excluded from the study. Questionnaires were provided to patients or their caregivers only after obtaining informed consent. The study was approved by the Local Ethical Committee of the Bambino Gesù Children’s Hospital of Rome.

### 2.2. Data Collection

#### 2.2.1. Demographic and Clinical Data

Patients recruited were re-evaluated for demographic and clinical data. Demographic variables included gender, age, and age at disease onset. Clinical features analyzed included the disability status, assessed using the Kurtzke Expanded Disability Status Scale (EDSS), an ordinal scale structured into eight functional systems based on neurological examination. Additional clinical parameters included disease duration, the number of days spent in the hospital during the year preceding recruitment—considering admissions for acute relapse treatment, chronic treatment administration, and clinical or instrumental assessments—and the relapse rate, calculated as the number of clinical relapses occurring in the year before recruitment. Information on disease-modifying treatments (DMTs) was also recorded, classifying therapies into high-efficacy agents (fingolimod, ocrelizumab, rituximab, and natalizumab) and moderate-efficacy agents (dimethyl fumarate, glatiramer acetate, and interferon beta) [26]. The total exposure time to DMTs was also considered.

Laboratory parameters were obtained from routine medical records, including cerebrospinal fluid (CSF) analysis from lumbar puncture performed at disease onset and blood tests conducted as part of the routine follow-up. CSF parameters included the presence of oligoclonal bands (OCBs), determined by isoelectric focusing combined with immunoblotting of matched serum and CSF sample pairs, the presence of pleocytosis, defined as a white blood cell count > 5 cells/mm^3^, and the IgG index in CSF.

The blood iron profile, assessed within two months of the interview for RLS, included plasmatic ferritin levels, serum iron levels, and hemoglobin levels.

Brain and spinal MRI scans were retrospectively reviewed, including those performed at disease onset and the most recent scans acquired within one year of recruitment. All MRI examinations were performed using a 3T scanner. Brain imaging sequences included axial and sagittal T2-weighted, FLuid-Attenuated Inversion Recovery (FLAIR)-weighted, T1-weighted, and contrast-enhanced T1-weighted sequences following gadolinium administration. Spinal MRI protocols included dual-echo (proton-density and T2-weighted) conventional and/or fast spin-echo sequences, STIR sequences as an alternative to proton-density weighted images, and contrast-enhanced T1-weighted spin-echo sequences in cases where T2 lesions were present.

Radiological data analysis considered the presence and number of lesions at each time point, the progression of lesion burden between the two scans, and lesion localization on T2-weighted and contrast-enhanced T1-weighted images, with specific attention to the cerebellum, brainstem, cervical cord, dorsal cord, and sacral cord.

#### 2.2.2. Questionnaire Structures and Interview Procedures

Participants meeting the inclusion criteria were asked to complete standardized questionnaires to screen for SDs and were interviewed by telephone to investigate symptoms suggestive of RLS.

##### Sleep Questionnaires

The questionnaires were administered in the morning at our POMS Center Clinic outpatient facility, during routine clinical follow-up for MS patients, or during the DMT infusion sessions with E.P., a clinician familiar with the patients.

For patients under 18 years of age, parents completed the Sleep Disturbance Scale for Children (SDSC) [27], while patients over 18 years old filled out the Pittsburgh Sleep Quality Index (PSQI) [28].

Participants were then categorized into two groups, “good sleepers” and “poor sleepers,” based on their SDSC and PSQI scores. “Poor sleepers” were defined as those who scored >5 on the global PSQI score or >39 on the total SDSC score.

##### RLS and Restless Sleep Disorder Investigation

Patients and their caregivers were interviewed by telephone to investigate symptoms suggestive of RLS. Interviews were conducted based on the latest age-appropriate criteria [15].

The severity of RLS symptoms was assessed using the IRLSSG rating scale (IRLS) [29], a widely used tool employed both on adults as well as on the pediatric population [19,30]. The IRLS consists of 10 items rated from 0 to 4, yielding a total score that categorizes symptom severity as mild (0–10), moderate (11–20), severe (21–30), or very severe (31–40) [29].

To better define the phenotype and differentiate RLS from restless sleep, we specifically investigated the presence of clinical criteria suggestive of RSD based on the 2020 recommendations [23], though the investigation was limited to clinical features as video-polysomnography (vPSG) was not performed (Table 2).

##### Quality of Life Investigation

Data about patients’ quality of life were collected through The Pediatric Quality of Life Inventory (PedsQL) [31], a health outcome measure designed for the pediatric population. It consists of 23 items divided into four scales: Physical, Emotional, Social, and School Functioning, where higher scores indicate better quality of life.

### 2.3. Statistical Analysis

Statistical analysis was performed using SPSS (Statistical Package for the Social Sciences) version 20. Data were presented as means ± standard deviation, medians, and percentages. Categorical data were analyzed using chi-square tests. The distribution of parametric data was assessed using Shapiro-Wilk tests. For comparisons of nominal data, *t*-tests were used. Pearson and Spearman correlations were employed to estimate relationships between sleep disturbance scores and clinical data. ANOVA was used for the comparison of demographic and clinical data between groups. A *p*-value of ≤0.05 was considered statistically significant.

## 3. Results

### 3.1. Demographic and Clinical Features

We screened 46 patients with POMS, of whom two were excluded due to unwillingness to participate. Consequently, 44 patients were enrolled in the study. All patients were diagnosed with the relapsing-remitting form of MS and were on DMT at the time of the study. Demographic and clinical features of the patients enrolled in this study are summarized in Table 3. The distribution of DMT types among the patients is shown in Table 4.

### 3.2. Sleep Questionnaires

A total of 41 fully completed sleep questionnaires were collected, with 26 patients (63.4%) completing the Sleep Disturbance Scale for Children (SDSC) and 15 patients (36.6%) completing the Pittsburgh Sleep Quality Index (PSQI). Based on the questionnaire scores, 16 patients (39%) were categorized as “good sleepers”, while 25 patients (61%) were categorized as “bad sleepers”. For details about questionnaire scores, see Appendix A.

Comparing demographic and clinical features (age, age at disease onset, disease duration, EDSS, days spent in the hospital, relapse rate, total exposure time to DMT) between the two groups, we found no significant differences. The “bad sleepers” had higher EDSS scores, but this difference did not reach statistical significance (*p* = 0.074) (see Appendix A).

We found no significant correlations between the type of DMT patients were on (categorized as High-Efficacy DMT and Moderate-Efficacy DMT) and the presence of SD (*p* = 0.45). Table 4 shows the distribution of DMT types between patients with and without SD.

There were no statistically significant correlations between quality of life scores and the presence of SD (R = 0.33, *p* = 0.23); although “bad sleepers” had lower quality of life scores, no significant differences were found comparing the “good sleepers” and “bad sleepers” groups (mean ±SD: 76 ± 16.6 vs. 66.8 ± 21.6, *p* = 0.21).

### 3.3. Restless Legs Syndrome (RLS)

All patients were assessed for the presence of RLS. Ten patients (22.7%) met the criteria for RLS. Of these, four patients (40%) had severe RLS, five patients (50%) had moderate RLS, and one patient (10%) had mild RLS. The demographic and clinical features of patients with and without RLS are presented in Table 5.

One patient with RLS had a positive family history of RLS, and five patients with RLS met the clinical criteria for RSD. A total of 85.7% of patients with RLS had pathological scores on the sleep questionnaires, with a significant correlation between the two factors (R = 0.53, *p* = 0.002).

The mean age at the time of assessment was significantly higher in POMS patients with RLS compared to those without (*p* = 0.028). Although the age at disease onset was higher in POMS patients with RLS compared to those without, this difference was not statistically significant. The mean EDSS score was significantly higher in the RLS+ group (*p* = 0.03). No significant correlation was found between the type of DMT the patients were on (categorized as High-Efficacy DMT and Moderate-Efficacy DMT) and the presence of RLS (*p* = 0.86). Table 4 shows the distribution of DMT type between patients with and without RLS.

There were no statistically significant correlations between quality of life scores and the presence of RLS (R = 0.32, *p* = 0.44), nor were significant differences found comparing “RLS+” and “RLS−“ groups (mean ±SD: 65.57 ± 24.9 vs. 70.9 ± 19.4, *p* = 0.53).

#### 3.3.1. RLS and MRI Features

MRI images were available for 43 patients at disease onset and 42 patients at the latest follow-up. The MRI findings for patients with and without RLS are shown in Table 6.

There were no statistically significant differences between the groups regarding MRI findings at either disease onset or the latest follow-up.

#### 3.3.2. RLS and Laboratory Features

There were no statistically significant differences or correlations between the groups based on laboratory data (IgG index, OCB positivity, Pleocytosis, Iron, Ferritin, and Hemoglobin levels), as shown in Table 7.

## 4. Discussion

Our study suggests a higher proportion of SDs in POMS patients (61%) compared to the general pediatric population (25%), based on the existing epidemiological data [32]. While data on the prevalence of SDs in POMS is limited, our findings align with studies on adult-onset MS, where SDs affect 42–65% of patients, approximately three times higher than in healthy individuals [6].

Previous research on SDs in POMS has yielded mixed results. For example, Zafar et al. [7] found no significant difference in sleep quality between POMS patients and healthy controls. Similarly, Stephens et al. [33], using actigraphy, reported no differences in sleep parameters such as sleep time, sleep efficiency, number of awakenings, number of minutes awake after sleep onset, sleep latency, or time in bed between these groups. Conversely, Jaeggi et al. [34] identified a higher risk of sleep-disordered breathing in youth with MS compared to controls. These discrepancies likely stem from differing methodologies and small sample sizes, emphasizing the need for larger studies with objective assessment tools.

Regarding RLS, we found a prevalence of 22.7% in the POMS population, consistent with previous reports of a higher prevalence of RLS in POMS compared to the general pediatric population (2–4%) [19,22]. Nearly all patients with RLS reported moderate-to-severe symptom severity and poor sleep quality, with a significant correlation between these factors. Furthermore, in our study, RLS patients were significantly older and had higher disability scores, in line with the findings of Yalcinkaya et al. [22].

Although the average EDSS score in both groups is clinically almost non-contributory, it is important to consider this result in the context that pediatric patients are less likely to exhibit overt clinical impairment, showing better recovery post-relapse compared to adults and accumulating less disability in the early years of the disease [35]. This is reflected in our sample, where most patients demonstrated either no clinical or only mild symptoms in a single functional domain. Moreover, it is well-established that incomplete recovery following the first relapse and the presence of clinical signs in the early years of the disease are negative prognostic factors for long-term disability accumulation [35,36]. Therefore, exploring this aspect longitudinally would be of great interest.

The high proportion of RLS in patients with MS, coupled with the association between greater disability and RLS in MS patients and the lower occurrence of familial RLS in these patients (10% versus 60% in idiopathic RLS [18]), could suggest that RLS in MS may be a secondary form linked to the underlying disease. Several studies support this pathogenetic hypothesis, based on RLS’s connection to lesions in motor control pathways, particularly the pyramidal system and spinal cord [37,38]. In fact, the pathophysiology of RLS in MS is thought to involve disruption of dopaminergic circuits, which may be further exacerbated by iron deficiency [38]. A key area of interest is the A11 diencephalo-spinal pathway, which is commonly affected by MS lesions [39]. Dopaminergic A11 hypothalamic cells, the major source of dopamine in the spinal cord, project widely throughout it and modulate pain and locomotor networks [38]. This hypothesis is reinforced by animal studies showing sleep-related limb movements in rats with spinal cord lesions [40] and increased locomotor activity following damage to the A11 pathway [41]. Furthermore, the A11 nucleus receives input from the suprachiasmatic nucleus, which regulates circadian rhythms, potentially explaining the fluctuation of RLS symptoms [41].

Iron deficiency is another key element in RLS pathogenesis, as it plays a crucial role in dopaminergic regulation. Low brain iron stores are commonly observed in RLS patients, and ferritin serum levels below 50 ng/mL have been identified as pathogenic in secondary forms related to iron deficiency [42]. The role of iron in RLS in MS remains controversial. Some studies have found increased iron deposits in brain tissues, while others report that serum ferritin levels remain normal in patients with stable MS but are elevated in those with chronically active disease, due to the inflammatory activation of ferritin as an acute-phase protein [20,43]. In our study, we found lower ferritin levels in the RLS group compared to the non-RLS group, though the difference did not reach statistical significance. Interestingly, a large portion of patients in the RLS group had ferritin levels below 50 ng/mL, and could be considered in the context of a potential link between iron deficiency and RLS in MS.

While studies in adult MS have found associations between the number or location of MRI lesions and RLS, our study did not show such a relationship [37,39]. This discrepancy between clinical and radiological findings is common for various symptoms in pediatric MS and may be due to more effective remyelination and lesion reversibility in younger patients [44].

It is also possible that neuroinflammation plays a more significant role in pediatric RLS compared to neurodegeneration [44]. OCB, IgG, and pleocytosis are well-established inflammatory markers in MS, correlating with disease severity and prognosis [45,46,47]. However, no studies have investigated neuroinflammatory status through these CSF markers in patients with both MS and RLS, despite evidence suggesting that alterations in other systems, such as the endogenous opioid and melanocortin systems, play a role in RLS pathogenesis [48]. Our study, in line with that of Yalcinkaya et al., did not find a statistically significant increase in CSF neuroinflammatory markers (OCB, IgG, and pleocytosis) in the RLS+ group. This suggests that in our sample, the inflammatory aggressiveness at disease onset may not be a decisive factor in the development of RLS. Unfortunately, we were not able to analyze CSF features closer to the time of the interview or include other markers employed to monitor disease severity and progression, such as neurofilament light chain. These findings highlight the need to broaden the analysis to identify more specific inflammatory biomarkers for RLS in MS.

In our sample, patients with RLS were significantly older compared to the group without RLS. Age is a known risk factor for RLS in both children and adults [18,19]. However, diagnosing RLS in younger children can be challenging due to difficulties in symptom description and incomplete reports from parents. Using age-appropriate tools, such as visual aids, can improve symptom description in younger children [49].

Differentiating RLS from RSD is an emerging challenge. Half of our patients with RLS also met the clinical criteria for RSD, suggesting a potential overlap in the pathogenesis of these conditions, possibly involving iron deficiency and disrupted dopaminergic pathways [23]. This finding warrants further investigation, given the limited literature on the topic.

This is the second study demonstrating the increased proportion of RLS in POMS patients compared to epidemiological data about the healthy pediatric population and its correlation with disability status.

Despite these interesting findings, our study has some limitations. We did not include a control group in our study, and we compared our results with the currently available epidemiological data, which could limit the validity and generalizability of our results, together with the relatively small sample size, affecting the ability to detect statistically significant differences in some analyses, such as correlations between RLS and blood iron profile, and not allowing the use of multivariate analyses, which would risk producing unreliable estimates. Additionally, the cross-sectional design does not allow for the determination of causal relationships between sleep disorders or RLS and other clinical symptoms or radiological parameters.

Data collection on sleep primarily relied on self-reported questionnaires, which may introduce response bias, particularly in pediatric patients who might struggle to accurately describe their symptoms. Moreover, we used two different sleep assessment tools, based on age validation, to ensure diagnostic accuracy; while this approach may introduce some heterogeneity that could affect data comparability, we tried to minimize this issue by reproducing the analyses within age-defined subgroups. The lack of objective tools, such as polysomnography or actigraphy, to assess sleep disorders is another limitation that could have reduced the accuracy of diagnosing certain conditions.

Finally, the variability in diagnostic criteria for RLS and other sleep disorders, combined with the challenge of distinguishing specific RLS symptoms from other pediatric conditions such as RSD, may have impacted diagnostic accuracy. Future studies should include larger samples, longitudinal methodologies, and objective diagnostic tools to enhance the accuracy and clinical relevance of these findings and to perform multivariable modeling to allow the identification of robust predictors of sleep outcomes.

## 5. Conclusions

Our study provides evidence of a higher proportion of SDs and RLS in pediatric MS patients compared to the general pediatric population, an area with limited existing literature. Future research should focus on exploring the MS characteristics most closely associated with RLS risk, utilizing longitudinal methodologies and objective assessments to clarify pathogenetic mechanisms and better understand the relationship between disease progression and RLS. Standardized methods for assessing SDs in POMS are needed to improve clinical detection, particularly in subthreshold cases not directly reported by patients, thereby enabling earlier intervention.

## Figures and Tables

**Table 1 jcm-14-03157-t001:** Factors to consider for the diagnosis of pediatric RLS.

The child can use his own words to report RLS symptoms. It is useful and efficient that the clinician assessing pediatric RLS is familiar with the terminology commonly used by children
The main factor that leads to the applicability of RLS diagnostic criteria is level of language and cognition, rather than chronological age
When assessing the presence of RLS, we should consider that impact on behavioral and educational functioning is more evident than on sleep and psychological aspects, as it is for adults
We can now refer to revised and simplified research criteria to diagnosticate probable and possible pediatric RLS
We should be aware that periodic limb movement disorder may manifest before the diagnosis of RLS

Abbreviations: RLS: Restless Legs Syndrome.

**Table 2 jcm-14-03157-t002:** Consensus diagnostic criteria for RSD.

All must be met: Restless sleep reported by the patient’s parent or caregiver
Restless sleep characterized by significant movements, occurring during sleep, involving large muscle groups of the entire body, including all four limbs, arms, legs, or head
The total movement index (by video analysis) through video-polysomnography should report five or more movements per hour of sleep
Restless sleep happens at least three times per week for at least three months
Restless sleep leads to significant daytime compromission in behavior, education, academics, social interactions, work, or other key areas of functioning
Restless sleep is not more appropriately accounted for by another sleep disorder, medical condition, mental health issue, behavioral disorder, environmental factor, or substance

Abbreviations: RSD: Restless Sleep Disorder.

**Table 3 jcm-14-03157-t003:** Demographic and clinical features of the study cohort.

Sex	
Male, n (%)	14 (31.8%)
Female, n (%)	30 (68.2%)
Age (y), mean (SD, range)	16.9 (3.2, 7.3–22.5)
Age at disease onset (y), mean (SD, range)	13.8 (2.8, 5.3–17.5)
Disease duration (m), mean (SD, range)	44.2 (31.1, 6–130)
Relapse rate, mean (SD, range)	0.1 (0.3, 0–1)
EDSS, mean (SD, range)	0.3 (0.6, 0–3)
Days per year spent in the hospital, mean (SD, range)	11.46 (5.2, 2–21)
Total exposure time to DMT, mean m (SD, range)	37.95 (26.4, 6–108)

Abbreviations: SD: Standard Deviation; y: years, m: months; EDSS: Expanded Disability Status Scale; DMT: Disease-Modifying Treatment.

**Table 4 jcm-14-03157-t004:** Distribution of DMT types between patients with and without SD, RLS.

	Total Patients	Good Sleepers	Bad Sleepers	RLS+	RLS−
	n. 44	(n.16, 39%)	(n.25, 61%)	(n.10, 22.7%)	(n.34, 77.3%)
**High Efficacy**					
Fingolimod	5 (11.3%)	2 (4.8%)	3 (7.2%)	1 (2.2%)	4 (9%)
Natalizumab	17 (38.6%)	4 (9.7%)	13 (31.6%)	4 (9%)	13 (29.5%)
Ocrelizumab	7 (15.9%)	3 (7.2%)	2 (4.8%)	2 (4.4%)	5 (11.3%)
Rituximab	7 (15.9%)	4 (9.7%)	2 (4.8%)	1 (2.2%)	6 (13.6%)
Tot	36 (81.7%)	13 (31.6%)	20 (48.6%)	8 (18.1%)	28 (63.5%)
**Moderate Efficacy**					
IFNb-1a	5 (11.3%)	1 (2.4%)	4 (9.7%)	1 (2.2%)	4 (9%)
Dimethyl Fumarate	3 (6.8%)	2 (4.8%)	1 (2.4%)	1 (2.2%)	2 (4.4%)
Tot	8 (18.1%)	3 (7.3%)	5 (12.5%)	2 (4.4%)	6 (13.6%)

Abbreviations: SD: sleep disturbance; RLS: Restless Legs Syndrome; IFNb-1a: Interferon b-1a; DMT: Disease-Modifying Treatment.

**Table 5 jcm-14-03157-t005:** Comparison of demographic and clinical features between patients with and without RLS.

	RLS+ (n.10, 22.7%)			RLS− (n.34, 77.3%)			*p*-Value
	Mean	SD	Range	Mean	SD	Range	
Age, y	18.9	1.5	16–22	16.3	3.4	8–22	**0.028**
Age at disease onset, y	15.1	1.6	12–17	13.4	3	5–17	0.093
Disease duration, m	46.5	32.2	8–108	43.5	35	6–113	0.79
Relapse rate	0.1	0.3	0–1	0.1	0.3	0–1	0.55
EDSS	0.7	0.1	0–3	0.2	0.6	0–3	**0.03**
Total exposure time to DMT, m	41.1	25.5	18–100	37	27	7–108	0.65

Abbreviations: SD: Standard Deviation; EDSS: Expanded Disability Status Scale; RLS: Restless Legs Syndrome; y: years; m: months; Restless Legs Syndrome; DMT: Disease-Modifying Treatment.

**Table 6 jcm-14-03157-t006:** Comparison of MRI findings between patients with and without RLS.

Onset MRI		RLS+ (n.10, 23.3%)	RLS− (n.33, 76.7%)	*p*-Values
N° T2W cranial lesions (mean, SD)		16.2 (18.4)	14.2 (13)	0.704
N° T1W-C cranial lesions (mean, SD)		4.3 (3.7)	4.1 (5.9)	0.956
N° T2W cerebellar lesions (mean, SD)		3 (4.4)	1.9 (3.5)	0.451
N° T2W brainstem lesions (mean, SD)		1.2 (1)	1.8 (2.2)	0.357
N° T2W spinal cord lesions (mean, SD)		3.1 (3)	2.2 (2.6)	0.407
N° T1W-C spinal lesions (mean, SD)		1.2 (1.6)	0.7 (1.3)	0.359
N° T2W cervical lesions (mean, SD)		1.2 (0.9)	1.2 (1.3)	0.926
N° T2W dorsal lesions (mean, SD)		1.6 (1.5)	1.1 (1.4)	0.381
N° T2W lumbar lesions (mean, SD)		0.3 (0.9)	0.2 (0.5)	0.812
**Latest MRI**		**RLS+ (n.10, 23.9%)**	**RLS− (n.32, 76.1%)**	***p*-values**
N° T2W cranial lesions (mean, SD)		19.5 (17.9)	17.4 (13.8)	0.703
N° T1W-C cranial lesions (mean, SD)		0	0	NA
N° T2W cerebellar lesions (mean, SD)		3.2 (4.3)	2.4 (3.6)	0.581
N° T2W brainstem lesions (mean, SD)		1.2 (1)	2.4 (2.6)	0.146
N° T2W spinal cord lesions (mean, SD)		3.8 (3)	3.2 (2.9)	0.58
N° T1W-C spinal cord lesions(mean, SD)		0.1 (0.3)	0	0.078
N° T2W cervical lesions (mean, SD)		1.4 (1)	1.6 (1.4)	0.6
N° T2W dorsal lesions (mean, SD)		2.1 (1.7)	1.5 (1.5)	0.31
N° T2W lumbar lesions (mean, SD)		0.3 (0.9)	0.2 (0.5)	0.83

Abbreviations: MRI: Magnetic Resonance Imaging; SD: Standard Deviation; RLS: Restless Legs Syndrome; T1W: T1-Weighted; T2W: T2-Weighted, T1W-C: Contrast-enhanced T1-Weighted; NA: not applicable.

**Table 7 jcm-14-03157-t007:** Comparison of laboratory features between patients with and without RLS.

	RLS+	RLS−	*p*-Value
IgG index (mean, SD)	0.95 (0.37)	0.91 (0.58)	0.9
Ferritin, ng/mL (mean, SD)	40.4 (33.7)	63.8 (50.8)	0.28
Iron, mg/dL (mean, SD)	75.8 (28.4)	74.4(26.68)	0.89
Hemoglobin, g/dL (mean, SD)	16 (4)	14 (3)	0.67
	**RLS+**	**RLS−**	**R, *p*-value**
OCB positivity (n°, %)	9 (26.4%)	25 (73.5%)	−0.24, 0.12
Pleocytosis (n°, %)	6 (85.7%)	20 (60.6%)	−0.16, 0.32
Ferritin < 50 ng/mL (n°, %)	5 (50%)	10 (29.4%)	0.21, 0.33

Abbreviations: OCB: Oligoclonal Bands; SD: Standard Deviation; RLS: Restless Legs Syndrome.

## Data Availability

The raw data supporting the conclusions of this article will be made available by the authors on request.

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
