# Peer review of "Sleep Disorders in a Sample of Patients with Pediatric-Onset Multiple Sclerosis: Focus on Restless Legs Syndrome"

_jcm, 2025, doi:10.3390/jcm14093157_

Round 1
Reviewer 1 Report
Comments and Suggestions for Authors
Methods: laboratory and MRI should be considered clinical data, and moved up to the first paragraph with demographics and clinical data.
it would be helpful to include dates of recruitment and the type of study (cross sectional cohort study)
I'm not understanding the "days in the hospital" variable - is this for their first relapse event, or what is the particular admission you are capturing?
Why were some POMS patients excluded? I understand the desire to not capture sleep within a relapse window, but could you have conducted the study after they recovered from relapse? Explaining this rationale would be critical. if you did exclude a small number, please put the total screened along with included POMS participants. Same with medications - you could adjust for these covariates in your analysis or stratify the cohort during your analysis, and sometimes it may inform you about the sleep pattern for these patients. If you chose to exclude based on medication, please also include how many you excluded.
Be careful with neuropsychological testing language. Neuropsychological testing implies cognitive testing. You did psychiatric screening, quality of life and fatigue, but no cognitive testing.
Results:
You do not need both means and medians in descriptive data - choose one based on normality of your data - either mean and SD or median and IQR
Elaborate on the type of DMTs the participants were on - high efficacy vs low efficacy, total exposure time to DMT
Demographic and clinical variables should be summarized in a table.
Table 4: check spelling. Not sure what midollar is? Did you mean medullar? Lombar is lumbar?
Please show the data for change in brainstem lesion load in tabular form showing the comparison between RLS+ and RLS- groups
Lab features: did you look at hemoglobin or iron levels in the RLS groups? Most POMS patients will have at least a CBC differential to tell you if they're anemic.
Also did you look at whether there's a difference in DMTs used between poor and good sleepers, and RLS+ and RLS- groups?
Table 5 - this data needs to be organized in a more digestable format. You are showing raw data on the beta and standard error for these scores. You talk about differences between good and poor sleepers in line 277, but this is not presented clearly in Table 5.
Table 6: you present median and SD. this should be mean and SD. Please check these numbers
for correlation, you need R as well as P value
Discussion: Your study is not a case-control, so I would be more conservative with the language about prevalence of sleep disorders in POMS compared to general pediatric. While this is very possible, you cannot appropriately ascertain this without a proper control. You can only suggest based on existing epidemiological data . For sleep disorders in pediatric population, it would be helpful to elaborate on the existing data - were they healthy patients recruited in primary care clinics, or were they evaluated in a sleep clinic in which there is some bias toward a higher proportion of sleep disturbance from various causes?
Line 313 - while you had a significant p value for EDSS differences, this is a difference between 0.3 and 0.7 which clinically is almost non-contributory on the EDSS (no symptoms vs functional score of 1 in one subscore). I would be careful on placing too much emphasis on this difference.
Line 337 - the presence of WBC, OCBs do not alone determine disease severity. other markers such as nFL are used to track disease severity and progression. Be careful wtih the over interpretation of these biomarkers. You have excluded participants with more active disease, so it's possible that you are missing an important subgroup of patients.
Line 367 - It is not surprising at all that fatigue is not correlated with RLS. This is a multi-faceted and often challenging symptom to tackle, and it's been shown that mental health plays an important role. Sleep is only one factor. MS affects all of these. I would rephrase your language here.
Line 378 - you cannot say prevalence without doing appropriate epidemiological analysis. You can say that you have observed a higher proportion of sleep disorders and RLS in POMS compared to the proportions reported in general pediatrics.
Line 382 - a mjor limitation is lack of control in this group to make meaningful comparisons in your observation.
Conclusion: you cannot say risk if you did not do a risk analysis (such as odd's ratio, etc). You did a correlation analysis.
Author Response
Please see the attachment.

|
Response to Reviewer 1 Comments
|
||
|
1. Summary |
|
|
|
Thank you very much for your time in reviewing this manuscript. Below, you will find our detailed responses, along with the corresponding revisions and corrections, which have been highlighted in the re-submitted files using track changes.
|
||
|
2. Questions for General Evaluation |
Reviewer’s Evaluation |
Response and Revisions |
|
Does the introduction provide sufficient background and include all relevant references? |
Yes |
We prefer to respond in a detailed and expanded manner in the point-by-point response letter below |
|
Are all the cited references relevant to the research? |
Yes/Can be improved/Must be improved/Not applicable |
|
|
Is the research design appropriate? |
Can be improved |
|
|
Are the methods adequately described? |
Can be improved |
|
|
Are the results clearly presented? |
Must be improved |
|
|
Are the conclusions supported by the results? |
Must be improved |
|
|
3. Point-by-point response to Comments and Suggestions for Authors |
||
|
Methods: laboratory and MRI should be considered clinical data, and moved up to the first paragraph with demographics and clinical data. Thank you for the suggestion, you can now find the paragraph in the manuscript under the Methods section- combined with demographics and clinical data. [Pg. 4, Par. 2.2.1, Line 127-147] it would be helpful to include dates of recruitment and the type of study (cross sectional cohort study). We appreciate this suggestion and have addressed it by specifying the study type at the beginning of the Methods section, in paragraph 2.1 [Pg. 3, Par. 2.1, line 100]. As for the recruitment dates, this information is provided in section 2.1, under “Subject recruitment” [Pg. 3, Par. 2.1, line 102]. I'm not understanding the "days in the hospital" variable - is this for their first relapse event, or what is the particular admission you are capturing? Thank you for highlighting this point. We have clarified this by adding details immediately after the mention of the variable. Our aim was to capture the overall extent of the patient's hospitalization, reflecting the total time spent in the hospital due to necessary admissions for the acute treatment of relapses, chronic treatment administrations, and clinical or instrumental assessments. This variable also takes into account the potential impact of prolonged hospitalizations – whether for disease reactivation or infusion therapies - on mental health, quality of life, and, consequently, sleep quality, in contrast to oral or subcutaneous therapies. [Pg. 4, Par. 2.2.1, line 119-121] Why were some POMS patients excluded? I understand the desire to not capture sleep within a relapse window, but could you have conducted the study after they recovered from relapse? Explaining this rationale would be critical. if you did exclude a small number, please put the total screened along with included POMS participants. Same with medications - you could adjust for these covariates in your analysis or stratify the cohort during your analysis, and sometimes it may inform you about the sleep pattern for these patients. If you chose to exclude based on medication, please also include how many you excluded. Thank you for this comment. What we meant is that we did not recruit patients during the acute relapse period, because the data would not have been consistent with the patient's baseline clinical condition, especially considering the likely use of high-dose corticosteroids, which significantly affect sleep quality. What we considered methodologically correct was to wait 30 days after the acute relapse, once the patient had returned to their baseline state, and recruit the patient at that time. We have revised the manuscript and added explanations to make the methodology clearer. As for medications, we only set the criterion to exclude patients on chronic treatment with drugs known to have a clear and significant impact on sleep. However, no such cases were encountered. Patients receiving corticosteroid therapy were only temporarily excluded and were interviewed after completing the treatment [Pg. 3-4, Par.2.1, line 105-109]. Be careful with neuropsychological testing language. Neuropsychological testing implies cognitive testing. You did psychiatric screening, quality of life and fatigue, but no cognitive testing. Thank you for pointing this out. Therefore, we have revised the language entitling “Psychological, quality of life and fatigue Questionnaires” [Pg. 5, line 161].
Results: You do not need both means and medians in descriptive data - choose one based on normality of your data - either mean and SD or median and IQR. Thank you for your suggestion. We have chosen to report the mean and standard deviation (SD) for descriptive data, as this best reflects the distribution of our variables. We have revised the text accordingly. Elaborate on the type of DMTs the participants were on - high efficacy vs low efficacy, total exposure time to DMT. We understand that this is an important clue that we missed. We have therefore revised the manuscript and add information about DMT in the paragraph dedicated to Demographic and Clinical features of the study cohort [Pg. 4, Par 2.2.1, line 122-126], and this information has been therefore included in the results section in the description of the sample [Pg, 6, Table 3, Pg. 9-10, Table 8]. We have furthermore analyzed any relation between the type of DMT the patient were on and the presence of sleep disturbance, Restless leg syndrome and fatigue [Pg. 9, Par. 3.5, line 296-302, Pg. 9-10, Table 8]. Demographic and clinical variables should be summarized in a table. Thank you for this suggestion, we have revised the paragraph dedicated to Demographic and Clinical features of the study cohort, summarizing data in the table following the paragraph [Pg. 6, Table 3]. Table 4: check spelling. Not sure what midollar is? Did you mean medullar? Lombar is lumbar? Thank you for bringing this typographical error to our attention. We have revised and corrected with “lumbar” and “spinal cord” [Pag. 7-8, Table 5]. Please show the data for change in brainstem lesion load in tabular form showing the comparison between RLS+ and RLS- groups. Thank you very much for this comment; it allowed us to reflect once again on our results. Considering that the correlation was statistically weak, qualitatively inconsistent, and not supported by any other MRI data, we believe its significance is negligible, or misleading. Therefore, we prefer not to consider it at the moment, taking the opportunity to pay particular attention to this matter and further investigate it during the patient follow-up process. Lab features: did you look at hemoglobin or iron levels in the RLS groups? Most POMS patients will have at least a CBC differential to tell you if they're anemic. We understand the importance of the role of iron profile, thus we added data about ferritine levels, iron levels, HB levels, comparing RLS + vs RLS- groups. We captured this data referring to blood exams performed near in time to the review for RLS presence. We have described this procedure in the Methods section [Pg. 4, Par 2.2.1, Line 133-134] and data about correlation with RLS in the result section [Pg. 8, Par. 3.3.2, Line 270-272, Pg. 8, Table 6] and discussed further [Pg. 11 Line 355-365]. Also did you look at whether there's a difference in DMTs used between poor and good sleepers, and RLS+ and RLS- groups? Again, we understand that this is an important clue that we missed. We have therefore revised the manuscript adding information about DMT compared between poor and good sleepers, and RLS+ and RLS- groups. You can find the results under the respective paragraphs [Pg.9, Par. 3.5, line 296-302, Pg. 9-10, Table 8] and discussed further [Pg. 12, Line 406-410]. Table 5 - this data needs to be organized in a more digestable format. You are showing raw data on the beta and standard error for these scores. You talk about differences between good and poor sleepers in line 277, but this is not presented clearly in Table 5. Thank you very much for pointing this out. We have rephrased the text [Pg.9, Par. 3.4.1, line 281-284] and simplified the table to improve readability [Table S3, supplementary information]. Since the table did not display statistically significant data, we decided to report the results in the text and move the table to the supplementary information, in order to make the reading of the text more fluid and highlight only the main information. Table 6: you present median and SD. this should be mean and SD. Please check these numbers for correlation, you need R as well as P value. Thank you for bringing this typographical error to our attention. We have revised and corrected with “mean” instead of “median”, and further add R values [Pg. 9, Table 7].
Discussion: Your study is not a case-control, so I would be more conservative with the language about prevalence of sleep disorders in POMS compared to general pediatric. While this is very possible, you cannot appropriately ascertain this without a proper control. You can only suggest based on existing epidemiological data. For sleep disorders in pediatric population, it would be helpful to elaborate on the existing data - were they healthy patients recruited in primary care clinics, or were they evaluated in a sleep clinic in which there is some bias toward a higher proportion of sleep disturbance from various causes? We agree with your comment, thus we rephrased the conclusion, explaining that “Our study suggest a higher proportion of SDs in POMS patients (61%) compared to the general pediatric population (25%), based on existing epidemiological data”, as reflected in the revised manuscript [Pg. 10, line 309-310]. We have specified that the 25% prevalence refers to the general pediatric population, which includes healthy children not seeking treatment for sleep issues or other conditions. We acknowledge that differences in study populations may influence prevalence estimates and have now clarified this limitation in the discussion [Pg. 12, Par.4, line 426-428]. Line 313 - while you had a significant p value for EDSS differences, this is a difference between 0.3 and 0.7 which clinically is almost non-contributory on the EDSS (no symptoms vs functional score of 1 in one subscore). I would be careful on placing too much emphasis on this difference. We appreciate your comment and have reviewed the manuscript to better clarify our findings. While the difference in the mean EDSS score between the two groups does not reach clinical significance, we believe it is still relevant, as it reflects a higher proportion of patients in the RLS group with minimal but present clinical signs, compared to the group without RLS, where most patients were completely asymptomatic. Given the negative prognostic implications of incomplete recovery from a relapse and the potential for disability accumulation in the early stages of the disease, we believe this data warrants consideration. [Pg. 10, Par.4, line 331-339] Line 337 - the presence of WBC, OCBs do not alone determine disease severity. other markers such as nFL are used to track disease severity and progression. Be careful wtih the over interpretation of these biomarkers. You have excluded participants with more active disease, so it's possible that you are missing an important subgroup of patients. Thank you for your insightful comment. We acknowledge that WBC and OCBs alone do not fully determine disease severity, and we have clarified in the text that other biomarkers, such as NfL, are also relevant in assessing neuroinflammation and disease progression, although they were not included in our analysis [Pg. 11, Par.4, line 380-383]. Regarding patient selection, we did not exclude individuals with more active disease; rather, we temporarily postponed their inclusion and reassessed them outside the acute relapse window, once they had recovered [Pg. 3-4, Par. 2.1, line 105-109]. Line 367 - It is not surprising at all that fatigue is not correlated with RLS. This is a multi-faceted and often challenging symptom to tackle, and it's been shown that mental health plays an important role. Sleep is only one factor. MS affects all of these. I would rephrase your language here. We agree with your comment, thus we rephrased our language [Pg. 12,Par.4, line 401-410]. Line 378 - you cannot say prevalence without doing appropriate epidemiological analysis. You can say that you have observed a higher proportion of sleep disorders and RLS in POMS compared to the proportions reported in general pediatrics. We agree with your comment, thus we rephrased our language [Pg. 12, Par.4, line 415]. Line 382 - a mjor limitation is lack of control in this group to make meaningful comparisons in your observation. We agree with your comment, thus we reviewed the limitations paragraph placing the lack of the control group as the main limitation of our study [Pg. 12, Par.4, line 426-428]. Conclusion: you cannot say risk if you did not do a risk analysis (such as odd's ratio, etc). You did a correlation analysis. We agree with your comment, thus we rephrased our language [Pg. 13, line 451]. |
||
Reviewer 2 Report
Comments and Suggestions for Authors
REVIEW
I have reviewed with interest the manuscript entitled “Sleep disorders in a sample of patients with Pediatric Onset Multiple Sclerosis: focus on Restless Leg Syndrome and impact on quality of life and psychological status” submitted for publication to Journal of Clinical Medicine.
The study aimed to evaluate the prevalence of SDs, particularly RLS, in a POMS cohort, and to examine associated features, risk factors, and impacts on quality of life and mental well-being.
Authors concluded an increased risk of SDs and RLS in pediatric MS patients. Clinical and neuroradiological data support a secondary form of RLS in this population.
I think this is an interesting article but it needs major changes, several points should be clarified by the authors, and some conclusions cannot be supported due to limitations. I have some comments to the author’s bellow.
Major concerns.
Abstract
line 18: sample 44 POMS patients
line 21&22: Three patients are missing? 16 good sleepers and 25 poor sleepers (total 41)
Line 25: the correlation may be due to the time of the evolution of MS. Please explain.
Introduction
The Introduction is too long, should be shortened and focused on the pediatric topic POMS. The information regarding the adult MS is too extensive.
Subjective sleep complaints and clinical variables such as fatigue, mood disturbances, evaluated by specific questionnaires have been included. An important cause of sleep alterations in some MS patients is nocturia - what is one of the causes that significantly disturbs sleep - and should be included together with its treatment.
I recommend to change the word cognitive impairment to “non motor MS symptoms”
Material and Methods.
The Expanded Disability Status Scale (EDSS) is not included in Methods and it deserves a little explanation.
Please provide a table listing MS drugs with beneficial, no or adverse effects on fatigue. This is very important for treating physicians and should therefore be detailed.
Please the information from the questionnaires (SDSC, PSQI, Fatigue Severity Scale, GAD- 7, PHQ-9) is unnecessary since the references are already given. This information should be eliminated.
Please provide alternative scales to access MS fatigue and highlight difficulties to differentiate between motor and cognitive fatigue.
Please specify where the questionnaires were conducted, by whom and at what time of the day.
Please detail if you have its hemoglobin and iron metabolism parameters (ferritin, transferrin saturation index and soluble transferrin receptor). These parameters are essential to determine whether the RLS is primary or secondary to MS.
Results.
Please the figures that are already shown in the tables must be removed from the text which make understanding easier.
RLS and MRI Features.
Line 270, please specify the level of the brainstem lesion that correlates with the presence of RLS, in other words, highlight the lesion location necessary to define an MS lesion as causal for RLS.
It would be interesting for the radiologist to review MRIs to assess brain iron if they have done any sequence that allows it.
Discussion.
Please discuss the dilemma that ferritin might be increased in autoimmune diseases even though not as a result of sufficient iron supply rather than a consequence of inflammation as ferritin is also an acute phase protein.
Conclusions.
The biggest limitation of this study is the absence of objective parameters obtained by a PSG recording. In this way, it is not possible to check whether the patients had PLMs what is the main cause of sleep fragmentation and disturbed nocturnal sleep in MS, in addition to nocturnal symptoms such as nocturia that has not been mentioned throughout the article.
Other important limitation is the lack of peripheral ion measurement. If we do not know if patients have iron deficiency It cannot be affirmed that RLS is secondary to EM.
The study brings some important clinical information, but the limitations mentioned above must be improved.
Comments on the Quality of English Language
I recommend checking the English by a native.
Author Response
Please see the attachment.

|
Response to Reviewer 2 Comments
|
||
|
1. Summary |
|
|
|
Thank you very much for your time in reviewing this manuscript. Below, you will find our detailed responses, along with the corresponding revisions and corrections, which have been highlighted in the re-submitted files using track changes.
|
||
|
2. Questions for General Evaluation |
Reviewer’s Evaluation |
Response and Revisions |
|
Does the introduction provide sufficient background and include all relevant references? |
Must be improved |
We prefer to respond in a detailed and expanded manner in the point-by-point response letter below |
|
Are all the cited references relevant to the research? |
Yes/Can be improved/Must be improved/Not applicable |
|
|
Is the research design appropriate? |
Must be improved |
|
|
Are the methods adequately described? |
Must be improved |
|
|
Are the results clearly presented? |
Must be improved |
|
|
Are the conclusions supported by the results? |
Must be improved |
|
Abstract line 18: sample 44 POMS patients line 21&22: Three patients are missing? 16 good sleepers and 25 poor sleepers (total 41). Thank you for highlighting this issue. The inconsistency in the numbers can be explained as follows: the total number of recruited patients was 44, for whom we had complete clinical data and data about RLS presence. Of these, 3 patients did not complete the sleep questionnaires, so we have available data for 41 patients regarding SD. These data and procedures are presented in the results section [Pg. 5, Par. 3.1, line 202 and Pg. 6, Par. 3.2, line 212]. Nevertheless, we understand that the presentation of the data in its current form is confusing, and therefore, we have revised the abstract accordingly [line 18 and line 23-24]. Line 25: the correlation may be due to the time of the evolution of MS. Please explain. As explained more extensively below, following a review of our results basing of your valuable comments, we prefer not to consider this data at the moment, because of the potential confounding factors and the weak robustness of the data.
Introduction The Introduction is too long, should be shortened and focused on the pediatric topic POMS. The information regarding the adult MS is too extensive. Thank you very much for your suggestion, we shortened and modified the introduction accordingly. Subjective sleep complaints and clinical variables such as fatigue, mood disturbances, evaluated by specific questionnaires have been included. An important cause of sleep alterations in some MS patients is nocturia - what is one of the causes that significantly disturbs sleep - and should be included together with its treatment. Thank you for pointing this out, we described nocturia as impacting sleep quality in the introduction [Pg,2, line 65-68]. I recommend to change the word cognitive impairment to “non motor MS symptoms”. We agree with your comment, thus we rephrased our language [Pg. 2, line 52].
Material and Methods. The Expanded Disability Status Scale (EDSS) is not included in Methods and it deserves a little explanation. We agree with your comment, thus we added a little explanation dedicated [Pg.4, Par 2.2.1, line 115-118]. Please provide a table listing MS drugs with beneficial, no or adverse effects on fatigue. This is very important for treating physicians and should therefore be detailed. Thank you for your valuable suggestion. We conducted a thorough literature review on the topic, but we found that the data are quite heterogeneous, influenced by factors such as follow-up duration, comparison with baseline fatigue rather than placebo or other DMTs, and the characteristics of the populations studied. Given that fatigue is a complex and multifactorial symptom, assessing it can be challenging, and providing raw percentages could be misleading. Moreover, several studies and meta-analyses do not show significant differences in fatigue levels when comparing different DMTs, especially when considering intracategorical comparisons (high-efficacy vs. moderate-efficacy treatments).
While we are unable to provide a detailed table listing the effects of MS drugs on fatigue due to these limitations in the available literature, we have taken your suggestion into account by providing a detailed discussion of the DMTs used in our sample and exploring any potential relationships with observed fatigue levels [Pg. 4, Par. 2.2.1, line 122-126; Pg. 6 table 3; Pg.9, Par 3.5, line 296-302; Pg.9-10, table 8; Pg.12, Par. 4 line 406-410] . We hope this approach addresses the importance of this topic while acknowledging the complexities involved. Please the information from the questionnaires (SDSC, PSQI, Fatigue Severity Scale, GAD- 7, PHQ-9) is unnecessary since the references are already given. This information should be eliminated. Thank you for taking this to our attention: we summarized main information about questionnaires and deleted concepts that can be found consulting the references. Please provide alternative scales to access MS fatigue and highlight difficulties to differentiate between motor and cognitive fatigue. We appreciate your comment and understand the importance of addressing both motor and cognitive fatigue in MS. While we acknowledge that the Fatigue Severity Scale (FSS) is not specifically validated for the pediatric population, we chose to use it in our study for its widespread use in pediatric samples, as supported by the existing literature. However, we recognize the limitations of the FSS, particularly its inability to distinguish between motor and cognitive fatigue.
In response to your suggestion, we also reviewed alternative scales for assessing fatigue in MS, such as the Modified Fatigue Impact Scale (MFIS) and the Pediatric Quality of Life Inventory (PedsQL) Fatigue Scale. These scales have been used in various pediatric MS studies, though they too present challenges in differentiating motor and cognitive aspects of fatigue. The difficulty in distinguishing between these two types of fatigue lies in their overlapping symptoms, with both types often being influenced by motor dysfunction, cognitive impairment, and mood disturbances, making it challenging to assess them separately with current tools.
We have added a discussion of these alternative scales and their limitations in the revised manuscript, acknowledging the complexity of assessing fatigue in pediatric MS and the need for more specific and sensitive tools in this area [Pg. 12, Par. 4, line 440-442]. Please specify where the questionnaires were conducted, by whom and at what time of the day. Thank you, we were happy to provide more detailed information about this topic; you can find it combined with respective paragraphs [Pg. 4-5, Par. 2.2.3, line 152-154, line 162-164]. Please detail if you have its hemoglobin and iron metabolism parameters (ferritin, transferrin saturation index and soluble transferrin receptor). These parameters are essential to determine whether the RLS is primary or secondary to MS. We understand the importance of the role of ferritine and iron profile, thus we added data about ferritine levels, iron levels, HB levels, comparing RLS + vs RLS- groups. We captured this data referring to blood exams performed near in time to the review for RLS presence. We have described this procedure in the Methods section [Pg.4, Par. 2.2.1, line 133-134] and data about correlation with RLS in the result section [Pg. 8, Par. 3.3.2, line 270-272; Pg.6, table 3] and discussed further [Pg. 11, Par.4, line 355-365].
Results. Please the figures that are already shown in the tables must be removed from the text which make understanding easier. Thank you, we have revised the paragraph dedicated to Demographic and Clinical features of the study cohort, summarizing data in the table following the paragraph and removing from the text [Pg.6, Table 3]. We trust that the data will become more accessible as a result of this modification. We did not perform the same operation about the section “Sleep Questionnaires” because the table related is not promptly available, since is collocated in the supplementary information [Table S2]. RLS and MRI Features. Line 270, please specify the level of the brainstem lesion that correlates with the presence of RLS, in other words, highlight the lesion location necessary to define an MS lesion as causal for RLS. Thank you very much for this comment; it allowed us to reflect once again on our results. Considering that the correlation was statistically weak, qualitatively inconsistent, and not supported by any other MRI data, we believe its significance is negligible, if not misleading. Therefore, we prefer not to consider it at the moment, taking the opportunity to pay particular attention to this matter and further investigate it during the patient follow-up process. It would be interesting for the radiologist to review MRIs to assess brain iron if they have done any sequence that allows it. Thank you for your suggestion, we appreciate this and could apply this analysis on further study. Discussion. Please discuss the dilemma that ferritin might be increased in autoimmune diseases even though not as a result of sufficient iron supply rather than a consequence of inflammation as ferritin is also an acute phase protein. Thank you for your suggestion; you can find this topic discussed along with our results on the iron profile of our sample [Pg. 11, Par.4, line 359-362].
Conclusions. The biggest limitation of this study is the absence of objective parameters obtained by a PSG recording. In this way, it is not possible to check whether the patients had PLMs what is the main cause of sleep fragmentation and disturbed nocturnal sleep in MS, in addition to nocturnal symptoms such as nocturia that has not been mentioned throughout the article. We comprehend your comment; indeed, we have mentioned among the limitations the lack of an objective method for the diagnostic formulation [Pg.13, Par.5, line 434-438, 442-444]. Nonetheless, the results regarding RLS, which follow exclusively clinical diagnostic criteria, are, in our opinion, relevant and supported by abundant and emerging literature. Considering these results, we are encouraged to continue and deepen our analysis using objective tools. Other important limitation is the lack of peripheral ion measurement. If we do not know if patients have iron deficiency It cannot be affirmed that RLS is secondary to EM. Thanks to the comments regarding this topic, we have gathered the information exposed in the paper and increased the analysis in this regard. The study brings some important clinical information, but the limitations mentioned above must be improved. Thank you for this comment, you are perfectly right. We modified and expanded the limitations section. |
||
Reviewer 3 Report
Comments and Suggestions for Authors
The authors in this paper describes well the correlations between POMS, sleep disturbances and rest leg syndromes and examine associated features, risk factors, and impacts on quality of life and mental well-being.
It would be useful to add whether the examined patients had cognitive deficits that could further influence SDs and RLSs and how they detect them.
In case of a negative response, on the contrary, it would be useful to add whether SDs and RLSs can influence cognitive deterioration in POMS.
Author Response
Please see the attachment.

|
Response to Reviewer 3 Comments
|
|
||
|
1. Summary |
|
|
|
|
Thank you very much for your time in reviewing this manuscript. Below, you will find our detailed responses, along with the corresponding revisions and corrections, which have been highlighted in the re-submitted files using track changes.
|
|
||
|
2. Questions for General Evaluation |
Reviewer’s Evaluation |
Response and Revisions |
|
|
Does the introduction provide sufficient background and include all relevant references? |
Yes |
We prefer to respond in a detailed and expanded manner in the point-by-point response letter below |
|
|
Are all the cited references relevant to the research? |
Yes/Can be improved/Must be improved/Not applicable |
|
|
|
Is the research design appropriate? |
Can be improved/ |
|
|
|
Are the methods adequately described? |
Yes |
|
|
|
Are the results clearly presented? |
Can be improved |
|
|
|
Are the conclusions supported by the results? |
Yes |
|
|
The authors in this paper describes well the correlations between POMS, sleep disturbances and rest leg syndromes and examine associated features, risk factors, and impacts on quality of life and mental well-being. It would be useful to add whether the examined patients had cognitive deficits that could further influence SDs and RLSs and how they detect them. In case of a negative response, on the contrary, it would be useful to add whether SDs and RLSs can influence cognitive deterioration in POMS.
Thank you for raising this important issue. Cognitive decline in MS, particularly in the pediatric population, has recently gained research interest, and we are currently conducting a longitudinal study on our POMS population. Unfortunately, the data regarding the cognitive level of the analyzed cohort are at the moment raw and incomplete. We have placed significant emphasis on highlighting the importance of assessing the presence of sleep disturbances, as these have a well-established indirect impact on cognitive functioning in POMS, mediated by their effect on psychological well-being and the perception of fatigue. While no longitudinal studies are currently available that directly assess the influence of sleep quality on cognitive decline in the POMS population, evidence in the adult MS population shows that individuals with sleep disturbances frequently report a decline in both self-perceived and objectively measured cognition. We were pleased to add a paragraph in the discussion section to further explore this topic [Pg. 12, Par.4, line 418-425]. |
|||
Round 2
Reviewer 1 Report
Comments and Suggestions for Authors
The authors did a great job addressing the questions raised on the first review. A few remaining minor edits:
- Exclusion of patients with relapse within 30 days: semantically, I would not consider this an exclusion criterion. Instead, this should be framed in the study design as an INCLUSION. You can say that you recruited participants with POMS diagnosis who are >30 days from acute relapse, etc... That way you don't need to mention that you re-recruited them after enrollment since that is confusing.
- DMT: I would reference table 8 in your clinical characteristics paragraph so the reader can refer to the later table to see breakdown of DMT
Author Response
Please see the attachment.

|
Response to Reviewer 1 Comments
|
||
|
1. Summary |
|
|
|
Thank you very much for taking the time to review this manuscript. Please find the detailed responses below and the corresponding revisions in track changes in the re-submitted files.
|
||
|
2. Questions for General Evaluation |
Reviewer’s Evaluation |
Response and Revisions |
|
Does the introduction provide sufficient background and include all relevant references? |
Yes |
We prefer to provide the corresponding response in the point-by-point response letter. Please see below. |
|
Are all the cited references relevant to the research? |
Yes/Can be improved/Must be improved/Not applicable |
|
|
Is the research design appropriate? |
Yes |
|
|
Are the methods adequately described? |
Can be improved |
|
|
Are the results clearly presented? |
Yes |
|
|
Are the conclusions supported by the results? |
Yes |
|
|
3. Point-by-point response to Comments and Suggestions for Authors |
||
|
||
Reviewer 2 Report
Comments and Suggestions for Authors
The authors sent a revised version of their article “Sleep disorders in a sample of patients with Pediatric Onset Multiple Sclerosis: focus on Restless Leg Syndrome and impact on quality of life and psychological status”.
They corrected all items proposed and discussed in detail questionable remarks.
I have no further remarks to make.
Author Response
Thank you very much for reviewing our work with such care and attention. Your comments have been invaluable in providing guidance to improve our work.